# The Coach–Athlete–Parent Relationship: The Importance of the Sex, Sport Type, and Family Composition

**DOI:** 10.3390/ijerph19084821

**Published:** 2022-04-15

**Authors:** Ausra Lisinskiene, Marc Lochbaum

**Affiliations:** 1Institute of Educational Research, Education Academy, Vytautas Magnus University, K. Donelaičio Str. 58, LT-44248 Kaunas, Lithuania; marc.lochbaum@ttu.edu; 2Department of Kinesiology and Sport Management, Texas Tech University, Lubbock, TX 79409-3011, USA

**Keywords:** youth sport, interpersonal relationships, psychological development, coaching, parenting

## Abstract

Interpersonal relationships exist in many forms within the sport environment. Athlete performance and career direction, at times, depend on their formed sport relationships. Positive and negative interpersonal relationships among the coach, the athlete, and the parent affects many athletes’ behavioral outcomes, such as continued participation. Our research aimed to understand whether the positive and negative processes in the coach, athlete, and parent interpersonal relationships depend on athletes’ sex, age, family composition, sport experience, and the type of sport. To achieve our research purpose, 632 volunteer student-athletes (aged 11–19) completed our survey. Our survey included the Positive and Negative Processes in the Coach–Athlete–Parent (PNPCAP) relationship scale and demographics (i.e., sex, age, family composition, years in competitive sport, and sport type). The study results revealed that positive processes, as measured by the positive PNPCAP subscale, were invariant to our categorical variables. However, participants’ self-ratings of negative PNPCAP-measured processes depended upon sex, sport type, and family makeup. Significant (*p* < 0.05) two-way interactions revealed boys involved in individual sports and residing without their parents or with one self-reported a higher level of the negative processes. The calculated effect size values with the other groupings were mostly medium in magnitude. The third significant two-way interaction resulted for sport type by family makeup. This two-way interaction revealed individual sport participants without or residing with one parent reported higher levels of negative processes. The effect size values were a mix of small and medium in meaningfulness. In conclusion, while positive Coach–Athlete–Parent processes appear invariant to our measured categorical variables, sex, sport type, and family makeup moderated the negative processes. Further research, such as mixed methods, is required to best understand and provide direction for intervention research to reduce negative processes in youth sport.

## 1. Introduction

Scientific research shows that youth sports is surrounded and interrelated with many social interactions (parents, coaches, athletes, peers, etc.), as well as emotions and behaviors, which can be seen as either positive (trust, communication, support, motivation, respect, cooperation, etc.) or negative (de-motivation, over-involvement, etc.) [1,2]. The most closely interrelated interactions with the athlete in youth sports associate with the coach and parent [1]. These three members of the sporting environment are the main players [1,2]. The significant scholarly focus has illuminated both positive and negative forms of sports coaching and sports parenting, from the perspectives of coaches [3,4,5,6,7,8], parents [9,10,11,12,13,14,15,16], and youth participants [16,17]. One less understood aspect, however, surrounds the potentially conflicting role of all three participants, namely the athletic triangle. The athletic triangle consists of the coach, athlete, and parent [1,2], and the relationships within this triad can have a significant impact on the psychological development of the athlete.

Interactions among coaches, parents, and athletes are well established and conceptualized [1,2]. Researchers have found that the interpersonal relationships between Coach–Athlete–Parent can be expressed either positively or negatively [1,2]. The level and degree of coach or parent involvement in youth sports can cause and affect the athletes’ sporting performance, motivation and mastery, and sporting career. From the parent perspective, various relationship dimensions exist in youth sports. For example, parents can be closely attached to a child versus de-attached [9,18,19], demonstrate autonomy/control behavior [9,10,15], parents can be involved/overinvolved/not involved at all in their children’s sporting activities. Sometimes, parents act as coaches and try to coach their child without trusting the coach. These parenting behaviors, however, impact children’s participation [16,17,18,19,20,21,22,23,24,25,26], continuation and motivation to remain in sports, and often, can “make” or “break” their children’s experiences [10].

Vukadinović and Rađević [11] provided the important insight that the athlete and parent relationship in adolescence weakens and the coach’s authority increases. This phenomenon can be explained because parent–child relations change dramatically when children step into teenage years [12]. For example, Lavallee and colleagues [13] indicated that the coach is the most important factor regarding social support with young athletes, and his role, besides parents’ and peers’ support, should be as the main initiator and motivator in any type of sport. Vukadinović and Rađević [11] indicated that any sport sets a high criterion of success for both coaches and athletes, which is the reason why the coach–athlete relationship should be crucial in achieving excellent results in sport. This relationship is particularly important in the age of adolescence, when the parent–athlete relationship is getting weaker and interpersonal relations and knowledge gained in that period of life, as a young athlete, are accumulating for a lifetime, whether they are positive or negative. Vukadinović and Rađević [11] noted that at the age of adolescence athletes usually spend much more time with coaches than with their parents, so the coach’s role in this period of a young athlete’s life is crucial and, therefore, it is very important that the coach is highly competent, educated, and skilled in his job with young ages.

The parent and athlete relationship, particularly in the individual sports context, could be explained through the work of Lisinskiene and her colleagues [1,2,9], where the parent and child attachment are of the highest importance. Bowlby’s attachment theory highlights that the quality of the early parent–child relationship becomes a basis of secure or insecure attachment forms in a child’s later years, especially the adolescent years. The importance of attachment becomes apparent in adolescence—the period of psychological and social transition from childhood to adulthood [14]. The early parent–child relationship plays a vital role in this period. Attachment and the relationship with parents change in the period of adolescence, when young people become more independent from their parents. In this period, the parent–child attachment weakens as young people are faced with new challenges; they seek independence and self-sufficiency, and develop their self-identity. Again, the researcher Lisinskiene [9] noted that secure attachment to family gives adolescents a more secure emotional basis that they can always rely on in the future and in any contexts.

As with coaches, many athlete relationships’ dimensions occur in youth sports. For example, coaches coaching philosophy can be autonomy versus controlling [3,5,9], coach-created coaching environment can be psychologically healthy or psychologically unhealthy and toxic [9,27], coach communication skills with an athlete can be open and honest versus private and reserved [27]. Likewise, parent or coach behaviors directly affect every athlete. Therefore, the athletic triangle receives attention in the scientific and academic context. Solid and healthy relationships between all three members in any sports context (individual or team sports) and for any athlete age or gender ensures a healthy youth sport environment. Most importantly, it provides maximum athlete enjoyment and involvement. There are several scientific studies regarding parenting in youth sports. Researchers have established and evaluated many of the parenting styles and behaviors in relation to youth sports [15], as well as coach involvement and behaviors [9,27]. However, to date, only Lisinskiene and her colleagues [1,2] have developed an instrument to measure coaches, parents, and athletes in one questionnaire, the Positive and Negative Processes in Coach–Athlete–Parent relationship (PNPCAP).

In a few publications, Lisinskiene and her colleagues [1,2] conceptualized the Coach–Athlete–Parent interpersonal relationships and developed the PNPCAP. However, a lack of studies that evaluate such interpersonal (triad) relationships from the perspective of athletes’ and regarding athletes’ age, gender, family composition, and sporting experience are nonexistent. Thus, a gap exists in the scientific literature of how these constructs affect the PNPCAP. Therefore, this research aims to understand whether the positive and negative processes in coach, athlete, and parent interpersonal relationships depend on athletes’ age, gender, family composition, sport type, and sporting experience. We forwarded no specific hypotheses given the newness of the PNPCAP, in relation to our measured constructs. It is logical that family makeup (residing with parents or not) could moderate the PNPCAP.

## 2. Materials and Methods

### 2.1. Participants

The sample consisted of 632 volunteer adolescents participating in a sport club at the time of data collection. Participants were in sport clubs, from the biggest towns in Lithuania, Kaunas, Vilnius, and Panevėžys with a minimum of three practices a week, engaged in competitive events, and within the sport club for at least a year. In all, 54.3% were female and 45.7% male and ranged in age from 11 to 19 (M age = 14.77 ± 2.32). Participants self-reported their home makeup in an open response format. Live with both parents and just one were the two most represented answers for data analyses. In addition to sex, age, and home life, participants self-reported characteristics concerning their sport participation. For the type of sport, the numbers were nearly identical, with 49.5% of athletes identified as participating in an individual sport and 50.5% in a team sport. Regarding sport experience, 12.4% reported having 2 years, 14.7% having 3 years, 15.2% having 4 years, 13.3% having 5 years, and 44.3% having 6 years or more of sporting experience.

### 2.2. Procedure

The lead researcher contacted several sports school administrators and explained the aims and the objectives of the research. Each sports school administrator received the approved ethical forms (approved by Vytautas Magnus University, ethical code SS4-01) to distribute to parents, the youths, and the entire sports school coaching staff. Once the approval process was completed, adolescent athletes received a web-based survey (Google Forms) link provided by sports school administrators and coaches.

### 2.3. Instruments

The survey comprised two main sections. The first section asked the potential participants to self-identify demographics and their sport playing, sex, age, family composition, type of sport, and sport experience. The second portion of the survey concerned their perceived interpersonal relationships in their current sport team. To assess their perceived interpersonal relationships, participants completed the Positive and Negative Coach–Athlete–Parent Interpersonal Relationships (PNPCAP) questionnaire developed by Lisinskiene and her colleagues [1,2]. The PNPCAP includes 11 items concerned with the entirety of interpersonal relationships considered for each question. The 11 items divide into two subscales, Positive C-A-P and Negative C-A-P, of which seven questions reflect positive relationship processes and the remaining four reflect negative processes. Positive scale items include topics such as support (e.g., to agree with and give encouragement), teamwork (the combined actions of the C-A-P working together effectively to achieve a goal), respect (politeness, honor, and care shown towards each other), and communication (sharing information). An example positive-scale question is “Mutual respect characterizes my C-A-P”. The negative scale includes topics such as overinvolvement (overstepping of boundaries), too demanding, and excessive expectations. An example negative-scale question is “In my C-A-P, at least one member expects too much”. All questions used a five-point Likert scale, varying from totally disagree (1) to totally agree (5).

### 2.4. Data Analyses

We performed all data analyses using SPSS version 26.0. (IBM, Chicago, IL, USA) and Intellectus Statistics (https://www.intellectusstatistics.com/). To understand the collected data and answer our main research questions, we first examined the PNPCAP data. Next, we used MANOVA with the two PNPCAP subscales as the dependent variables with each of our categorical variables (sex, age, family composition, sport type, and sport experience) to gain an understanding of potential group effects before testing for interactions. To determine whether group main effects resulted, we examined the univariate F-tests for each PNPCAP subscale. For categorical variables of more than two, we used the Tukey HSD follow-up test to locate potential differences. Though interactions precede group effects, we examined all possible interactions up to many possible three-way interactions. We planned to plot significant interactions and calculate effect size differences inside of running potentially numerous *t*-tests. Regarding meaningfulness of differences, we examined partial eta-squared (η^2^) following Cohen’s [28] guidelines of 0.01 (small), 0.06 (medium), and 0.14 (large) and if needed calculated Hedges’ *g* using Cohen’s [24] guidelines of 0.20 to 0.39 (small), 0.40 to 0.79 (medium), and greater than 0.80 (large).

## 3. Results

### 3.1. Overview of PNPCAP Items, Subscales, and MANOVA Results

Table 1 contains the PNPCAP items and subscale data. Participants self-reported higher scores for the positive CAP items (Q1–Q7) relative to the scale and to the negative CAP items (Q8–Q11). The standard deviation units and 95% confidence interval widths were smaller for the positive CAP items. More variability existed in the negative CAP items. An examination of the kurtosis and skewness data demonstrated all items were acceptable as normally distributed. Concerning the two subscales, the reliability coefficients were acceptable at 0.85 (positive CAP subscale) and 0.81 (negative CAP subscale). The subscales differed to a large degree (paired samples *t*(631) = 25.39, *p* < 0.001, *g* = 1.65) and were moderately correlated (*r* = −0.36).

### 3.2. Group Analysis Statistics for the Positive and Negative PNPCAP Subscale

Initially, we ran separate MANOVA analyses with the two subscales as the dependent variables and the categorical variables as the fixed factor. Two significant MANOVA analyses resulted. The first was for sport type, Wilks’ Lambda F(2, 629) = 4.71, *p* < 0.01, η^2^ = 0.02, and the second for family makeup, Wilks’ Lambda F(4, 644) = 3.81, *p* < 0.01, η^2^ = 0.02. Table 2 contains the PNPCAP positive subscale data analyses for our five categorical variables. Though two of the MANOVAs were significant, none of the univariate F statistics were significant at the traditional level (i.e., *p* < 0.05) for the positive CAP subscale for any of our categorical variables. Given this result, we did not conduct any other analyses on the positive CAP subscale.

As Table 2 contained all information for the PNPCAP positive subscale analyses, Table 3 contains the PNPCAP negative subscale analyses. Though interactions supersede group main effects, as mentioned in our data analysis plan, we sought to first gain insight on any possible group differences. Two univariate F statistics, sport type and family makeup, were significant at the traditional level, with partial eta-squared, suggesting small meaningful differences. Examination of the data per categorical indicated individual compared to team sport participants reported a higher value on the negative CAP subscale. This difference was small in meaningfulness. Participants residing with neither parent scored higher on the negative CAP subscale than those residing with one parent, Tukey HSD without parents greater than with one parent mean difference = 0.41 (95% CI 0.00, 0.82), SE = 0.17, *p* < 0.05, *g* = 40.

### 3.3. Group Interactions for the Negative PNPCAP Subscale

Table 4 contains the PNPCAP items and subscale data. As several possible interactions existed, we examined all possibilities, stopping at three-way interactions (e.g., sex by sport type by family makeup) for the negative PNPCAP subscale. A few significant (*p* < 0.05) two-way interactions resulted. The first (see Figure 1) two-way interaction was for sex by sport type, F(1, 314) = 4.18, *p* = 0.03, η^2^ = 0.01. Boys who self-identified as playing an individual sport reported a higher level of negative PNPCAP subscale score compared to the other sex by sport type groups. The effect size differences were all medium in meaningfulness compared to individual-sport boys to team-sport boys (*g* = 0.62), individual-sport girls (*g* = 0.59), and team-sport girls (*g* = 0.60).

The next significant two-way interaction (see Figure 2) resulted for sex by family makeup, *F*(2, 314) = 3.89, *p* = 0.02, η^2^ = 0.02. On inspection of the data, they suggested examining differences among boys without parents and with one parent to the other groups. The effect size values were a mix of small and medium in meaningfulness, as referenced to boys residing without parents: boys residing with both parents (*g* = 0.49), girls residing without parents (*g* = 0.38), girls residing with one parent (*g* = 0.71), and girls residing with both parents (*g* = 0.32). Effect size values referenced to boys residing with one parent were at least small in meaningfulness, as follows: both residing with both parents (*g* = 0.32), girls residing without parents (*g* = 0.21), and girls residing with one parent (*g* = 0.52).

The last significant (*p* < 0.05) two-way interaction concerned sport type by family makeup, *F*(2, 314) = 2.91, *p* = 0.05, η^2^ = 0.02. When examining Figure 3, individual sport participants without parents and with one parent self-reported higher levels of negative PNPCAP scores. Effect size values referenced to individual sport participants residing without parents were as follows: individual sport residing with both parents (*g* = 0.40), team sport residing without parents (*g* = 0.25), team sport residing with one parent (*g* = 0.71), and team sport residing with both parents (*g* = 0.31). Effect size values referenced to individual sport participants were as follows: individual sport residing with both parents (*g* = 0.41), team sport residing without parents (*g* = 0.25), team sport residing with one parent (*g* = 0.76), and team sport residing with both parents (*g* = 0.30).

## 4. Discussion

This research aimed to understand whether the positive and negative processes in coach, athlete, and parent interpersonal relationships depended on number of athlete characteristics and sport type. Most of our interaction results, all with the negative PNPCAP subscale, concerned family composition. It seems living without one’s parents, or with just one, impacts boys in individual sports more than girls in team or individual sports. The exact reason for this is unknown. Harwood and colleagues’ [29] study highlighted that the motivational climate, for example, in a competition setting for an individual sport athlete, remains largely understudied as well. Harwood et al. [29] also stated that such an understanding, through appropriate measurement, appears needed within individual sports, given that parents frequently interact with their child before, during, and after competition. This is an important contextual point about individual sports and the reliance on parental support [29].

Studies on the psychological climate of sports have revealed that parental behaviors have an impact on determining positive outcomes in the development of youth talent, at various levels [17,18,19,20]. Perhaps without one’s parents, or with only one, missing or different behaviors are present. According to Van de Pol et al. [30], athletes in individual sports report higher levels of ego orientation in competition than team-sport athletes. Likewise, based on Van de Pol and colleagues’ research [31], athletes in individual sports demonstrated higher ego orientation in competition than training. Such goal-related sensitivities in individual sport competition settings render the climate around these individuals worthy of more precise and careful investigation [29]. The family composition is important to the athlete and if one parent is not present or the family makeup is not stable, it can affect the athlete’s sporting experience, success, and continuation of sport [9]. Through the qualitative phenomenological analysis [32], we found that for individual-sport athletes, both parents’ involvement is very important to the athlete because individual sport is different from team sports, where the interaction between the members and the coach has a higher intensity.

Based on Lisinskiene’s [9] work, this phenomenon is explained regarding gender differences as well. Lisinskiene’s research showed that boys had higher scores in parent trust compared to girls. This is an important insight regarding gender differences and the individual context of sport. Boys are more likely to accept parents’ feedback and trust them compared to girls. It might be that the girls more emotionally accept or reject the feelings and the relationship with parents. In addition, Lisinskiene’s work and in-depth interviews with athlete adolescents revealed that parental involvement in children’s sport is more important in the early period of sporting life and becomes less appreciable or unwelcome when children gain sporting experience. In the background of parent–child interrelations in sporting activities, adolescents’ alienation from parents is more common than communication and trust.

The alienation of older adolescents from parents in sports has several causes: growing children seek to be independent of their families in sports, and some parents behave inappropriately in youth sport and embarrass their children. Interviews with athlete adolescents revealed that some parents demonstrate substandard behavior in relation to their children, having no loyalty towards other participants of the sporting activity. However, financial support remains an important factor of parental support, at all levels of sporting experience. The degree and form of parental involvement in children’s sport influence the effectiveness of parent–child educational interaction. The degree and form of involvement chosen by the parents are not always appropriate and encouraging, and not always acceptable to adolescents. Therefore, Lisinskiene and colleagues [1,2] developed the C-A-P concept, where coaches, athletes, and parents need to be seen as one direction, interact, and understand each other’s expectations, to be involved in a positive way. Both coach and parent play the most important role in an athlete’s life [1,2].

Regarding age differences, the researcher in [9] states that active participation of all three members of the athletic triad (coaches, athletes, and parents) is important and necessary for positive development throughout adolescence (early, middle, and late adolescence). As family involvement is important in any athletes’ age or stage, it is important to note that in the very early years of a child’s involvement in the sports context, family relationships are crucial. For example, the researchers in [33] developed a unique intervention program that involved the participation in sports of all the parents and children. The program integrated psychological, educational, and sports skills into pre-organized sports training sessions. The study results showed that the intervention program that lasted one year had a positive impact on the overall family relationships and communication, and parent–child relationship in the sports context. The intervention program suggests that parental involvement in the intervention program positively affected parent–child attachment, the quality of interpersonal relationships between the parent and the child, and effective parenting strategies. Last, Turman [34] examined parental and family influences that help encourage young athletes’ sport participation. Turman reported that the dimensions of family sport orientation served as significant predictors for each compliance dimension, while athlete and parent sex differences emerged for activation of impersonal commitment and rewarding behaviors.

The foundation of Knight and Holt’s theory [35] helps to explain the optimal family involvement and its importance in youth sports. The theory states that healthy family relationships within the sporting environment is achieved when parents both understand and enhance their child’s experiences. The theory highlights that the recognition of each athlete’s ability and individuality play the most important role in athletes’ positive development through sports.

In summary, research shows [1,2,9] that parent participation is of the highest importance in athletes’ lives and this participation is welcomed by and is necessary for the athlete. However, athletes state that parent involvement, as well as coach training philosophy, should be positive, motivated, supported and not controlling. Active participation of all three members of the athletic triad (coaches, athletes, and parents) is important and necessary for the positive development of this system. Such a three-dimensional educational system may be more effective if parents were more actively and positively involved in the sport of their children, so that athletes are not pressed but motivated by the psychologically positive environment; coaches are given continuous learning possibilities. The harmonization of the athletic triad remains the main challenge of further research on this issue.

## 5. Limitations and Ideas for Future Research

As with all cross-sectional studies, limitations exist in our work. Such questionnaire work requires participants. Convenience sampling introduces biases and, in our data, those biases are unknown. It could be an over- or under-sampling of any one of our variables relating to all possible combinations (sex by team sport by family composition, etc.). Our second limitation concerns not knowing how many participants began and stopped completing our questionnaire. Last, we evaluated the PNPCAP and the categorical variables through difference statistics. The found relationships raise more questions than answers. Hence, we suggest a few future research suggestions. 

Very recently, Lochbaum and his colleagues [36] published a systematic review of sport psychology and performance meta-analyses. In an examination of the 30 included meta-analyses, one finds a few cohesion and performance meta-analyses in sport, e.g., Ref. [37]. The PNPCAP has much promise to expand the cohesion and sport performance literature to the impact of cohesive interpersonal coach, athlete, and parent relationships on sport performance. Researchers should place the PNPCAP in the context of performance, akin to the cohesion literature with sport performance. Another viable line of research concerns a better understanding of the interaction of sex, sport type, and family makeup. In this case, mixed methods (integrating quantitative and qualitative study designs) are a positive future direction. Last, longitudinal and even intervention research to gain an understanding of how the PNPCAP relates with many often-studied sport psychology constructs, such as moods and emotions [38], achievement goals [39,40], and climate [40,41] will be of great value. Moreover, longitudinal research should incorporate sport enjoyment and performance.

## 6. Conclusions

The PNPCAP is a new measure in youth sports. Our research, in a large sample, demonstrated that the positive PNPCAP subscale is invariant to often-measured categorical or demographic variables. This result eases concerns regarding a research agenda into this subscale. However, interactions resulted for the negative PNPCAP subscale; thus, providing clear future research directions for youth sport at all levels, grassroots to professionalized, exist in all countries. Hence, understanding and correcting negative perceptions of the Coach–Athlete–Parent interpersonal relationship can only improve the youth sport experience.

## Figures and Tables

**Figure 1 ijerph-19-04821-f001:**
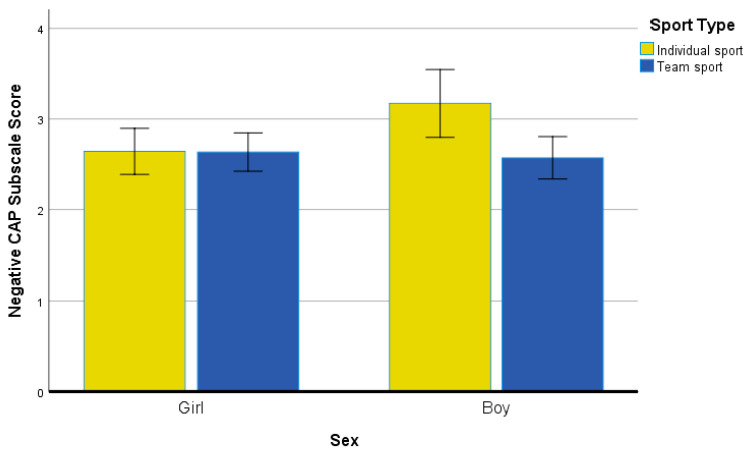
Negative CAP Subscale Scores for Sex and Sport Type Groups.

**Figure 2 ijerph-19-04821-f002:**
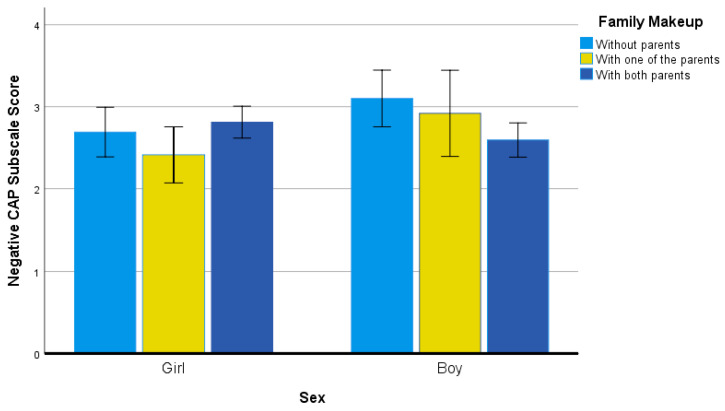
Negative CAP Subscale Scores for Sex and Family Makeup Groups.

**Figure 3 ijerph-19-04821-f003:**
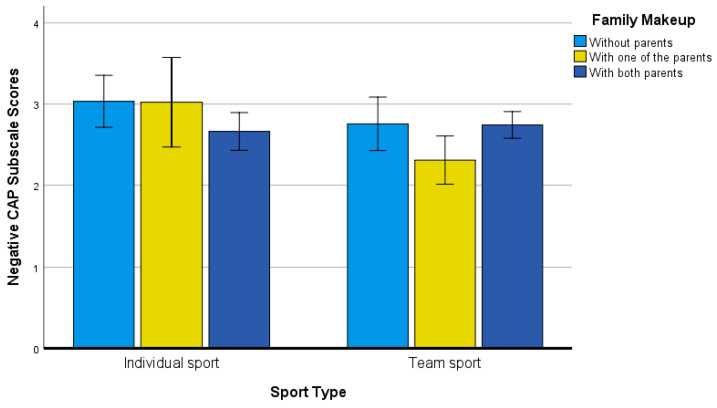
Negative CAP Subscale Scores for Sport Type and Family Makeup Groups.

**Table 1 ijerph-19-04821-t001:** Summary statistics for each of the PNPCAP questions and the total score for the two subscales.

Variable	M	SD	95% LL	95% UL	Min	Max	Kurtosis	Skewness
Q1: Reliable during hardship	3.94	0.98	3.86	4.01	1.00	5.00	0.39	−0.87
Q2: Are a team	4.00	0.89	3.93	4.06	1.00	5.00	0.77	−0.83
Q3: Is positive	4.04	0.81	3.98	4.11	1.00	5.00	1.29	−0.89
Q4: Works together	3.94	0.90	3.87	4.01	1.00	5.00	0.59	−0.80
Q5: Mutual respect	4.06	0.83	4.00	4.12	1.00	5.00	1.10	−0.90
Q6: Is supportive	4.10	0.76	4.04	4.16	1.00	5.00	1.08	−0.77
Q7: Listens to each other	3.99	0.85	3.93	4.06	1.00	5.00	0.67	−0.75
Q8: Expects too much	2.62	1.10	2.53	2.71	1.00	5.00	−0.63	0.26
Q9: Oversteps boundaries	2.54	1.13	2.45	2.62	1.00	5.00	−0.57	0.42
Q10: Too demanding	2.86	1.19	2.76	2.95	1.00	5.00	−1.01	0.07
Q11: Over involved	2.82	1.17	2.73	2.91	1.00	5.00	−0.80	0.16
Positive CAP subscale	4.01	0.63	3.96	4.06	2.00	5.00	−0.32	−0.25
Negative CAP subscale	2.71	0.92	2.64	2.78	1.00	5.00	−0.46	0.11

Note. 95% LL = 95% confidence interval lower limit; 95% UL = 95% confidence interval upper limit; Positive CAP Q1 to subscale = Q1–Q7; Negative CAP subscale = Q8–Q11.

**Table 2 ijerph-19-04821-t002:** Group difference statistics by categorical variables for the PNPCAP positive subscale.

Categorical Variable	Groups	n	M	SE	95% CI		Difference Statistics		
					LL	UL	Univariate F	*p*	η^2^
Sex	Girl	343	4.01	0.03	3.94	4.08			
	Boy	289	4.01	0.04	3.93	4.08	F(1, 631) = 0.01	0.94	0.00
Sport type	Individual	313	4.04	0.04	3.97	4.11			
	Team sport	319	3.97	0.04	3.91	4.04	F(1, 631) = 1.94	0.17	0.00
Family Makeup	No parents	70	4.11	0.08	3.96	4.26			
	One parent	57	3.93	0.09	3.76	4.10			
	Both parents	199	3.95	0.05	3.86	4.04	F(2, 323) = 1.81	0.17	0.01
Age group	11–13 yrs.	220	3.97	0.04	3.88	4.05			
	14–16 yrs.	255	4.00	0.04	3.92	4.07			
	17–19 yrs.	157	4.09	0.05	3.99	4.19	F(2, 629) = 1.88	0.15	0.01
Sport experience	2–3 yrs.	216	3.95	0.04	3.86	4.03			
	3–4 yrs.	163	4.00	0.05	3.90	4.10			
	5–6+ yrs.	253	4.07	0.04	3.99	4.14	F(2, 629) = 2.05	0.13	0.01

Note. n = number of participants; M = mean; SE = standard error; CI = confidence interval; LL = 95% confidence interval lower limit; UL = 95% confidence interval upper limit; η^2^ = partial eta-squared.

**Table 3 ijerph-19-04821-t003:** Group difference statistics by categorical variables for the PNPCAP negative subscale.

					95% CI		Difference Statistics		
Categorical Variable	Groups	n	M	SE	LL	UL	Univariate F	*p*	η^2^
Sex	Girl	343	2.70	0.05	2.60	2.80			
	Boy	289	2.72	0.05	2.61	2.82	F(1, 631) = 0.04	0.84	0.00
Sport type	Individual	313	2.78	0.05	2.68	2.88			
	Team sport	319	2.64	0.05	2.53	2.74	F(1, 631) = 4.13	0.04	0.01
Family makeup	Both parents	199	3.95	0.05	3.86	4.04			
	No parents	70	2.85	0.12	2.62	3.08			
	One parent	57	2.43	0.13	2.18	2.69			
	Both parents	199	2.72	0.07	2.59	2.86	F(2, 323) = 2.98	0.05	0.02
Age group	11–13 yrs.	220	2.63	0.06	2.51	2.75			
	14–16 yrs.	255	2.74	0.06	2.63	2.86			
	17–19 yrs.	157	2.76	0.07	2.61	2.90	F(2, 629) = 1.18	0.31	0.00
Sport experience	2–3 yrs.	216	2.75	0.06	2.63	2.87			
	3–4 yrs.	163	2.78	0.07	2.64	2.92			
	5–6+ yrs.	253	2.63	0.06	2.51	2.74	F(2, 629) = 1.72	0.18	0.01

Note. n = number of participants; M = mean; SE = standard error; CI = confidence interval; LL = 95% confidence interval lower limit; UL = 95% confidence interval upper limit; η^2^ = partial eta-squared.

**Table 4 ijerph-19-04821-t004:** Interactions of interest for the PNPCAP negative subscale.

Interaction Groups		n	M	SE	95% CI		Difference Statistics		
					LL	UL	Univariate F	*p*	η^2^
Girl	Individual sport	166	2.64	0.13	2.39	2.90			
	Team sport	177	2.64	0.11	2.42	2.85			
Boy	Individual sport	147	3.17	0.19	2.80	3.55			
	Team sport	142	2.57	0.12	2.34	2.81	F(1, 314) = 4.47	0.03	0.01
Girl	Without parents	40	2.69	0.15	2.39	2.99			
	With one of the parents	36	2.41	0.17	2.07	2.76			
	With both parents	107	2.81	0.10	2.62	3.01			
Boy	Without parents	30	3.10	0.18	2.75	3.45			
	With one of the parents	21	2.92	0.27	2.39	3.45			
	With both parents	92	2.59	0.11	2.39	2.80	F(2, 314) = 3.89	0.02	0.02
Individual sport	Without parents	37	3.04	0.16	2.71	3.36			
	With one of the parents	15	3.02	0.28	2.47	3.58			
	With both parents	66	2.66	0.12	2.43	2.90			
Team sport	Without parents	33	2.76	0.17	2.43	3.09			
	With one of the parents	42	2.31	0.15	2.01	2.61			
	With both parents	133	2.74	0.08	2.58	2.91	F(2, 314) = 2.91	0.05	0.02

Note. n = number of participants; M = mean; SE = standard error; CI = confidence interval; LL = 95% confidence interval lower limit; UL = 95% confidence interval upper limit; η^2^ = partial eta-squared.

## Data Availability

All data are available from the corresponding author.

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
