# Peer review of "The Coach–Athlete–Parent Relationship: The Importance of the Sex, Sport Type, and Family Composition"

_ijerph, 2022, doi:10.3390/ijerph19084821_

Round 1
Reviewer 1 Report
Dear Authors,
I am enclosing my review report.
Kind regards

Reviewer 2 Report
I find the revised study interesting and innovative because it analyses and provides data on a subject that needs to be studied in depth and also uses different independent variables that help to better explain a complex phenomenon such as interpersonal relationships in the field of sport.
The research requires some minor revisions, which I will explain below:
- It would be advisable for the summary not to end in the results section.
- From my point of view, the instrument should be better explained, especially the subscales.
- The discussion could be extended, as the results provided are broad and rich to be discussed, comparing them with others and suggesting explanations as to why these evidences. There is a lack of citations in the discussion with respect to each of the independent variables.
- Finally, there are a number of formal issues that need to be addressed. For example:
- The references
- Dot after the bracket in the text (e.g. in line 35: of coaches [3-8], parents [9-11]).
- Dots instead of commas inside tables
Reviewer 3 Report
Introduction
Background should be improved. Authors state some studies investigated something but do not provide what the studies have found so we understand the importance of the study.
Page 2 of 14, line 61: I suggest making a new paragraph from here until the end of the introduction.
Finish the aim sentence with (.) not with (?). Add hypotheses after the aim.
Methods
Participants
Page 2 of 14, line 84 (M age = 14,77 +/- 2,326): Should the +/- be ±? Also, 2,326 could be rounded to 2 decimal places like in the M values.
Procedure
In my experience with questionnaires, participants usually do not fully understand each question or they misunderstand some of them. How did you control or attempted to control the understanding of each question in the questionnaire?
If you just distributed questionnaires, the possibility of misunderstanding of some questions should be added in limitation section.
Instrument
Instrument needs to be explained in more details. Later, in results you are using variable names and groups. This should be clearly explained in methods. It would be useful to understand all this before the results and discussion sections so the results and discussion are more smooth. I think adding the instrument as supplement file could be helpful to readers. If that is possible, I suggest adding it.
Add the reliability statement for this instrument (Cronbach alpha) calculated on your sample.
Results
Check the Table formatting. You did not format all tables the same way. Check the number formatting in tables. Either use (,) or (.) for decimal places. Report results before the table rather than after (when possible).
The first paragraph of results section explains the variables. I suggest explaining this in methods section so you can state your results clearly with additional explanations what is what.
Table 1 - you have Respect two times. Add notes explaining C>A&B and C>A
Table 2 – you have Respect two times. Add notes explaining A>B, and A >B&C
Table 3 – Add notes to explain abbreviations like in Tables 1 and 2. In this table you have some number after A>B. Why is this different here than in the previous two tables?
Table 4 - you have Respect two times. Add notes to explain abbreviations like in Tables 1 and 2. In this table you have some number after A>B. Why is this different here than in the previous two tables?
Figure 1 – I think the figure title should be below the figure and it could be more informative. You should mark the significance on figure. Like this, I do not quite understand what do you refer to when you state that something is different here.
Page 7 of 14, line 167 – should be show instead of shows. In addition, the first two sentences could be combined to one: “Figure 1 shows that adolescents who live with both parents have Positive processes that appear more often compared to adolescents who live without parents or one of the parents”.
Page 7 of 14, line 168: you state “In contrast, we can see negative C-A-P processes in those who live with both parents more often than with one parent or without parents”. Considering absolute numbers this statement is right but we can also see that the number of subjects with both parents is larger. Could you provide proportions of subsamples providing negative C-A-P processes?
Page 7 of 14, line 170-171: “This figure explains and visualizes the table 2, 3 and 4 study results in more detail”. This figure as it is now does not provide more information. Adding proportion differences and significance would provide meaningful insights.
Figure 2 – you already have mean values in Table 5 so you can omit this figure as it is only a distraction.
I suggest removing correlation analysis as it does not add anything to the aim of this study.
Discussion
Discussion does not discuss the results nor compare them with previous studies. Conclusion is short and do not provide implications for practice.
Be more concise. Avoid inserting sentences that do not mean anything in relation to your results and study goal and hypotheses.
Page 12 of 14, line 5-7: “First, age differences were evaluated concerning PNPCAP. We evaluated the overall adolescence period (early adolescence, middle and late adolescence period). In this case, we have divided the adolescence age stage into three age groups (11-13, 14-16, 17-19)”.
This should be explained in methods section and no need to repeat here. Discuss the meaning of your results and compare to other studies.
Page 12 of 14, line 10: Omit bracket in front of (,).
Page 12 of 14, line 14-15: “Age differences were evaluated in more depth and the family composition concerning adolescents' age was highlighted”.
I do not see the point of this sentence.
Page 12 of 14, line 14-15: “Regarding age differences researcher [9] states that active participation of all three members of the athletic triad coaches, athletes, and parents is important and necessary for the positive development throughout adolescence age (early, middle, and late adolescence age)”.
This statement is too general. Are there any implications regarding the results that were explained by this statement?
Page 12 of 14, line 22-26: You only state what could be observed in results. You need to build on this. What is the meaning of this? What are the finding s of other studies on this? What are the implications for practice?
Page 12 of 14, line 27-32: Same as previous comment.
Page 12 of 14, line 48-49: You state: “In this sense, we can conclude that the higher the sporting experience the more positive C-A-P relationship processes appear”.
Your results also showed that older participants showed higher Positive C-A-P than younger participants. The question arises whether the Positive CAP related to age or the length of sporting experience. You should discuss this and provide insights.
References
There are 24 references. It seems too low for this complex topic. I believe authors should build better background and stronger discussion by referencing other studies that cover this topic.
Reviewer 4 Report
Interpersonal relationships within the Coach-Athlete-Parent triangle are of high interest and practical meaning. I would suggest however, a clearer presentation of the results and a more in-depth interpretation of the findings. The authors should provide explanation to the statistical results. For example, why older adolescent athletes had higher scores in positive scales than younger athletes and what does this mean in coaching practice? Is there any connection with training demands? One would believe that with the increase in training demands and the need for success in youth sport older athletes would be under higher pressure increasing their scores in negative scales (e.g. overinvolvement). Also, there is no explanation for the controversial results of the athletes living with both parents (higher scores in both positive and negative scales).
In the presentation of the results, I would suggest keeping only the main elements of the questionnaire (5 scales).
Abstract
The abstract includes only a presentation of the significant differences, without referring to any conclusions.
Grammar concerns in line 21, please check (…did not had…)
Please, delete numbers from keywords
Introduction
A brief description of the elements included in the questionnaire would give a better understanding of the results. For example, what does respect, or support mean in an athletic context, and how the perception of these elements change across age? Does support mean the same during early and late adolescence?
Please, check the format of the in-text citations. It seems to me that in lines 40 and 64 the surname of the first author is missing, which should be followed by the reference number in square brackets.
Methods
Please describe inclusion and exclusion criteria. Were the participants registered athletes of athletic clubs? Were there any criteria regarding minimum involvement? (e.g. training hours/week). It is reasonable to think that the degree of involvement in sports (competitive vs recreational) or the competition level (elite vs subelite) would change the way the athletes perceive their sporting activity and the attitude of their coaches and parents.
Please, include the sport disciplines and how you classified sports as individual/team sports.
How did you define sporting experience?
Ln84-87: Grammatical concerns, please check. What do you mean that athletes identified as having 2 years of sporting experience? If the athletes had 2 years of sporting experience, then this should be mentioned (e.g. 12.4% had 2 years of sporting experience). If this is uncertain, then please specify.
Please use a period for decimal points instead for a comma throughout the entire manuscript also including the tables.
Ln89: grammatical issues, please check (…Prior to…). Please correct to….researchers contacted...
Please provide the ethical approval number and specify the Institute of the Ethics Committee.
Ln106: Please check the simultaneous use of and and etc within the same sentence.
Results
The manuscript includes many tables, which is hard to follow. I suggest presenting only the main 5 elements of the questionnaire. I don’t understand the reason to have duplicated results within the same table. For example, table 1 includes the same results for ’Mutual respect characterizes my C-A-P’ and for ’P_respect’. It is redundant information and should be removed.
The title of the tables sounds unprecise. Please, use consistent titles for all tables. Table 1 contains the descriptive results for PNPCAP scores by age group. Formatting of the tables is not consistent. Please, use the same formatting for all tables.
Please, use maximum 2 decimals for all results.
On the pages containing tables, it seems that the right end of the text is missing. Please check and correct.
The results concerning age and sporting experience are misleading. Older athletes may have higher sporting experience, and this is absolutely an expected result. However, this should be considered when examining groups of different sporting experience, otherwise it may result in interpretation bias. I suggest examining the effects of sporting experience within the same age group, thus removing the influence of age.
Less experienced athletes scored higher on over-involvement scale, how can this be explained?
Athletes living with both parents had higher scores in both positive and negative scales. This looks controversial and deserves more discussion.
Figure 2 looks confusing and hard to understand. Again, I suggest keeping and presenting only the most important results. Differences in the various scales are not consistent! In most cases, athletes with 2 years experience had lower scores compared to the other groups, indicating that positive attitude increases with the increase in training background, I think you need to better highlight this result.
Discussion
The discussion repeats the results and there is no true interpretation of the results. You need to explain the outcomes of the statistical analysis. A simple report of the significant differences between sub-groups is not enough. What is the relevance of the presented findings? How can these findings be transferred to the athletic context? How can these findings be integrated to the relevant literature? These aspects have not been addressed adequately in the discussion.
Question mark should be removed from the first sentence.
Ln9: this sentence is not clear, please rephrase.
Ln10: this sentence is not clear, please rephrase.
Ln11-12: this sentence is not clear, please rephrase.
Ln27-31: in the first sentence of this paragraph the authors refer to the differences between sub-groups of various sporting experience; however, in the next sentences the type of sport (individual/team sports) is mentioned.
Ln33: grammatical concerns, please check (…there were found several statistical…)
Ln40: as above
Ln55: the authors state that parents should be more actively involved in the sport of their children. This should be explained in more details. What does parents’ involvement mean? Is there any evidence to support that increased parents’ involvement is beneficial for the sporting activity of their children? A major issue in today’s youth sports is the excessive involvement of the parents, disturbing in many cases coaching activity and eventually negatively influencing the development of the children. A more thorough analysis is needed here, supported also by previous findings from the literature.
As mentioned before there is no discussion at all for the controversial results of athletes living with both parents.
Round 2
Reviewer 4 Report
The revised manuscript has been improved significantly, particularly the presentation of the results. With some minor corrections it could be a valuable contribution to the literature. Still, I recommend another review round. It seems to me that the most interesting finding of this study appears in the figures (1, 2, and 3) indicating that boys and individual sport participants seem to be more affected by family status demonstrating higher scores in negative scales. This deserves some more analysis and discussion and could be the take home message here. By addressing this point and correcting some grammatical issues, it could be considered for publication.
Author Response
Dear Reviewer,
Thank you very much for your valued comments and suggestions. We have attached our response to your comments.
Thank you,
Authors
